# Comment on Zaher-Sánchez et al. The Management and Prevention of Delirium in Elderly Patients Hospitalised in Intensive Care Units: A Systematic Review. *Nurs. Rep.* 2024, *14*, 3007–3022

**DOI:** 10.3390/nursrep15020035

**Published:** 2025-01-24

**Authors:** José Lucas Sena da Silva, Juliana Caldas

**Affiliations:** 1Postgraduate Department, Bahiana School of Medicine and Public Health, Salvador 40290-000, BA, Brazil; caldasjuliana@gmail.com; 2Hospital São Rafael, Rede D’Or, Salvador 41253-190, BA, Brazil

**Keywords:** delirium, elderly, prevention

## Abstract

**Background/Objectives**: The systematic review by Zaher-Sánchez et al. evaluated the effectiveness of pharmacological and non-pharmacological interventions in preventing and managing delirium in elderly patients admitted to intensive care units (ICUs). This commentary discusses its importance, limitations, and dilemmas regarding delirium prevention. **Conclusions**: The need for effective measures for preventing delirium is imperative, and the findings to date support the benefits of non-pharmacological interventions, but we still do not have a sufficient body of evidence to choose specific interventions.

We found the article by Zaher-Sánchez et al. [1] to be insightful and timely. Their systematic review evaluated the effectiveness of pharmacological and non-pharmacological interventions in preventing and managing delirium in elderly patients admitted to intensive care units (ICUs). Although this study is of paramount importance and highlights the dilemmas regarding delirium prevention, several limitations affect the interpretation of their conclusions.

Initially, the data from the review should be assessed with caution, considering the heterogeneity of study designs, sample sizes, and demographic characteristics. For instance, the studies vary in age inclusion criteria, ranging from individuals over 18 years to those over 75 years. This aspect is particularly significant for evaluating the profile of adverse effects of pharmacological interventions, given the unique characteristics of elderly populations in terms of pharmacokinetics and pharmacodynamics. Additionally, the variety of interventions, often represented by individual studies, introduces limitations. The data were also not subjected to pooled analysis, which could have provided additional insights regarding efficacy, heterogeneity, and consistency across the studies.

Second, the authors’ findings indicated that the effectiveness of these strategies has yet to be consistently demonstrated. Notably, they found that dexmedetomidine showed potential benefits in reducing the incidence of delirium compared to olanzapine, although it was associated with a higher incidence of adverse events. In addition, melatonin and valproic acid also showed some promise in reducing delirium. In this context, delirium is likely to become an increasingly frequent area of concern in the intensive and surgical care of elderly patients, especially considering the global trend in population aging. However, elderly patients are particularly susceptible to the risks of polypharmacy, which raises concerns regarding the use of pharmacological approaches for delirium prevention. It is important to highlight that, despite their common use in the ICU, certain drugs carry notable risks, such as dexmedetomidine, which is associated with bradyarrhythmia, and antipsychotics, which carry a risk of arrhythmias and extrapyramidal effects [2,3,4]. Given these considerations, prioritizing non-pharmacological interventions appears to be a prudent strategy for delirium prevention.

Another important point is that the findings in this systematic review also support the benefits of non-pharmacological interventions for delirium prevention, consistent with previous systematic reviews [5]. Current evidence suggests that non-pharmacological interventions may be equally or even more effective than the pharmacological ones. A range of non-pharmacological, unimodal, and multimodal interventions were studied, and, to date, none proved to be superior. A meta-analysis conducted by Janssen et al. (2019) [5] showed that multicomponent interventions can successfully reduce the incidence of delirium, along with the use of antipsychotics, bispectral index-guided anesthesia, and the administration of dexmedetomidine during anesthesia. Considering the multifactorial etiology of delirium and the accumulated evidence regarding the benefits of multimodal non-pharmacological interventions, it seems plausible to prioritize a multimodal intervention at this time. The National Institute for Health and Care Excellence, for example, provides guidelines for delirium prevention, suggesting a tailored multicomponent intervention package delivered by a trained multidisciplinary team [6]. Nevertheless, the current literature remains limited, with a lack of high-quality studies and meta-analyses in this area, so we still lack studies on the superiority of a specific non-pharmacological intervention.

Furthermore, several risk prediction models for delirium were investigated; however, robust studies comparing the performance of these tools are still lacking, with few meta-analyses that compared various strategies, identifying two tools as seemingly more promising in terms of predictive accuracy: PREdiction of DELIRium in ICu patients (PRE-DELIRIC) and early PRE-DELIRIC (E-PRE-DELIRIC) [7,8]. Nonetheless, the authors of these meta-analyses highlighted the small sample sizes, the inconsistency in defining some of the predictive criteria among different tools, variations in study characteristics, divergent assessments of the delirium outcome, and data loss, negatively impacting consistency and heterogeneity. Alongside the development of effective alternatives for delirium prevention, the impact of stratifying strategies is also of great importance, especially for healthcare systems, which should utilize their resources efficiently. In alignment with this perspective, recent studies analyzed the effectiveness of utilizing a risk prediction tool for postoperative delirium. These studies demonstrated that the implementation of a risk prediction tool, combined with non-pharmacological interventions, resulted in a reduction in postoperative delirium and one study also suggested that it led to shorter hospital stays [9,10].

In conclusion, the rational use of resources in an aging population, which are increasingly required in hospital care, requires consistency and a robust evidence base to identify the most effective interventions for delirium prevention. In resource-limited hospitals and countries, implementing multiple interventions can be costly, highlighting the importance of prioritizing those with the greatest impact. Although we have evidence regarding this efficacy, we cannot yet assert that a more effective strategy exists, nor do we have assessments of non-inferiority among non-pharmacological strategies, which should be of essential scope in future studies. Further research with greater methodological rigor is essential to confirm the information we have so far. Additionally, it seems beneficial for the study of optimal preventive strategies to also seek predictive tools for delirium risk, enabling future research to shed light on populations more vulnerable to the development of this disorder. Nevertheless, it can be asserted that the implementation of non-pharmacological interventions for delirium prevention should be included in nursing care guidelines, as they provide not only delirium prevention and potentially other outcomes but also promote a patient-centered approach and enhance the patient’s experience.

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
