# Peer review of "Comment on Zaher-Sánchez et al. The Management and Prevention of Delirium in Elderly Patients Hospitalised in Intensive Care Units: A Systematic Review. Nurs. Rep. 2024, 14, 3007–3022"

_nursrep, 2025, doi:10.3390/nursrep15020035_

Round 1

Reviewer 1 Report

Comments and Suggestions for Authors

The commentary offers a thoughtful critique of Zaher-Sánchez et al.'s article and effectively highlights critical issues regarding delirium prevention in elderly ICU patients. However, by incorporating more specific examples, addressing methodological limitations, and providing a stronger call to action, the commentary could significantly enhance its scholarly contribution and practical implications for nursing and healthcare practice.

Strengths:

  1. Timeliness and Relevance: The commentary effectively underscores the importance of addressing delirium in the context of an aging population and the increasing prevalence of ICU admissions among the elderly.
  2. Balanced Perspective: The critique acknowledges both the potential benefits and risks of pharmacological approaches, such as the findings on dexmedetomidine, melatonin, and valproic acid, while emphasizing concerns about polypharmacy and adverse events.
  3. Advocacy for Non-Pharmacological Interventions: Highlighting non-pharmacological strategies as viable and potentially superior alternatives to pharmacological measures aligns with a growing body of evidence and offers a practical, patient-centered approach.

Suggestions:

  1. Specificity in Limitations: While the commentary notes that the current literature lacks validated predictive tools and high-quality studies, it could expand on how these limitations affect clinical practice and research priorities. For instance, examples of gaps in predictive tool validation or a brief mention of promising but underdeveloped tools would provide more depth.

  2. Critique of Methodology: The commentary could delve deeper into the methodological limitations of the systematic review by Zaher-Sánchez et al. For example, were there biases in study selection, or were certain populations underrepresented? Addressing these aspects would enhance the critique's scholarly rigor.
  3. Implication to Nursing: The commentary could conclude with a more explicit call to action, emphasizing the need for further research, improved predictive tools, and the integration of non-pharmacological interventions in clinical guidelines in nursing. This would help position the commentary as not only a critique but also a forward-looking contribution to the field.

Reviewer 2 Report

Comments and Suggestions for Authors

Dear authors,

The abstract introduces the topic well, highlighting the importance of non-pharmacological interventions in preventing delirium. It is well-organized into sections that address limitations, results, and practical implications. The commentary provides a balanced evaluation of the original article’s conclusions, pointing out both advances and gaps.

Some suggestions for improvement include:

  • Identifying which non-pharmacological interventions are the most promising in terms of cost-effectiveness. Previous studies are mentioned, but the argument could be strengthened by summarizing their key findings.
  • Exploring in greater depth the specific risks of the pharmacological interventions mentioned, such as hypotension and arrhythmias.
  • Clarifying recommendations for future research or practical implementation of non-pharmacological interventions, as well as prioritizing interventions, especially in resource-limited settings.

Reviewer 3 Report

Comments and Suggestions for Authors

Thank you for the commentary, which is a timely submission overall. This commentary may be improved given some points listed below. Major point: the authors should provide more references and evidence given their second point about polypharmacy. Minor point: Please proofread for minor grammatical errors, such as “delirium elderly” (should be “delirium in elderly patients”).

Round 2

Reviewer 1 Report

Comments and Suggestions for Authors

The authors have provided clear and comprehensive point-by-point responses to the reviewers' concerns. Each comment has been thoughtfully addressed, and the revisions made to the manuscript are appropriate and align with the feedback provided.

The revised manuscript reflects improvements in clarity, organization, and content, ensuring that the concerns raised by the reviewers have been resolved. The authors' efforts to incorporate these changes enhance the overall quality of the work.

Based on the responsiveness and thoroughness demonstrated, the manuscript is ready for acceptance, pending any final editorial adjustments.

Reviewer 2 Report

Comments and Suggestions for Authors

Dear Authors,

Congratulations on the revisions, which provide greater support to the article.